# Emergence of Hierarchical Layers in a Single Sheet of Self-Organizing Spiking Neurons

**Paul Bertens**
Department of Brain and Cognitive Engineering
Korea University
Seoul, South Korea
paulbertens@korea.ac.kr

**Seong-Whan Lee**
Department of Artificial Intelligence
Korea University
Seoul, South Korea
sw.lee@korea.ac.kr

## Abstract

Traditionally convolutional neural network architectures have been designed by stacking layers on top of each other to form deeper hierarchical networks. The cortex in the brain however does not just stack layers as done in standard convolution neural networks, instead different regions are organized next to each other in a large single sheet of neurons. Biological neurons self organize to form topographic maps, where neurons encoding similar stimuli group together to form logical clusters. Here we propose new self-organization principles that allow for the formation of hierarchical cortical regions (i.e. layers) in a completely unsupervised manner without requiring any predefined architecture. Synaptic connections are dynamically grown and pruned, which allows us to actively constrain the number of incoming and outgoing connections. This way we can minimize the wiring cost by taking into account both the synaptic strength and the connection length. The proposed method uses purely local learning rules in the form of spike-timing-dependent plasticity (STDP) with lateral excitation and inhibition. We show experimentally that these self-organization rules are sufficient for topographic maps and hierarchical layers to emerge. Our proposed Self-Organizing Neural Sheet (SONS) model can thus form traditional neural network layers in a completely unsupervised manner from just a single large pool of unstructured spiking neurons.

## 1 Introduction

Neurons are highly organized in the brain such that neurons encoding similar features are clustered closer together [1, 2]. These clusters of neurons form large hierarchical modules along the cortex. There is the formation of the major distinct cortical regions (visual cortex, auditory cortex, motor cortex etc.), and also subdivisions within each region (e.g. V1, V2, V3, etc.) [1]. Furthermore these regions are divided into cortical columns [3], where nearby cortical columns also encode similar features. For example neurons encoding edges of similar orientations are clustered together in the visual cortex, forming so called topographic maps [4, 5, 6]. These topographic maps have been extensively studied and are found everywhere along the cortex [1, 2].

Classically deep neural networks (DNNs) are constructed by stacking multiple layers on top of each other, manually designing the neural architecture [7]. Recent techniques in neural architecture search do try to find more optimal architectures for a given problem [8, 9], but they still consist of modules of layers that are combined to optimize some objective function. This creates an inherent bias in the type of information the network can process, and each architecture has to be optimized for the task at hand. In this paper a different approach is taken instead, we define local learning rules such that a neural architecture of multiple hierarchical regions automatically emerges. This enables us to start with a large pool of completely unstructured neurons that self-organize into logical modules.

36th Conference on Neural Information Processing Systems (NeurIPS 2022).

An important distinction should also be made between cortical regions and "layers" in relation to biological networks. Although Convolutional Neural Networks (CNNs) [7] consist of stacking multiple layers of neurons on top of each other, these layers are actually closer to the behaviour of cortical regions like V1, V2 etc. where each subsequent region increases in abstraction level. There are also layers in biological neural networks (i.e. the brain), generally divided into 6 groups (Layer I to Layer VI), but they do not serve the same function as layers in CNNs [10, 11]. Within this paper we use the layer terminology as used in CNNs, and treat them to be equivalent to the cortical regions in biological brains (see also figure 1).

## 1.1 Related work

Several spiking models exist to model biological spiking neural networks (SNNs) [12, 13, 14]. One of the simplest is the integrate-and-fire model (IAF) [15], which sums up the incoming input over time and fires a spike when reaching some threshold. More advanced approaches also exist, like the Hodgkin-Huxley [16] and Izhikevich model [17], which can exhibit more diverse behaviour like bursting [18] and spike frequency adaptation [19]. Although spiking neural networks are still computationally more expensive relative to DNNs, they are biologically more realistic and allow for extremely sparse processing [20]. They can also perform synaptic weight updates that take into account the timing of each individual spike through spike-timing-dependent plasticity (STDP) [21, 22, 23].

There has also been previous works on the formation of neural topographic maps [24, 25, 26]. One of the most common methods to form such maps are Self-Organizing Maps (SOMs), which are capable of producing a low dimensional representation of the input, mapping similar concepts closer together [27]. These are typically trained by randomly initializing a single layer artificial neural network (ANN) [28], and then matching a new input example to the unit that has the highest response (winner-take-all). This winner-take-all approach however has the limitation of only allowing a single unit to be active in the entire layer. They also can only form a single layer at a time. Combining both self-organizing maps and spiking neural networks has also been explored before [29, 30, 31]. These can form realistic topographic like maps as found in the visual cortex [4]. Hierarchical maps can also be created by manually stacking multiple maps. However, the main limitation is still that the layer architecture has to be predefined, which is what is universally done in both spiking and standard neural networks.

## 1.2 Contributions

The main motivation behind the work in this paper is the observation that cortical regions (i.e. layers) are formed on a 2-dimensional sheet instead of through vertical stacking as done in CNNs [1]. This means neurons have to wire together laterally while maintaining the same spatial mapping. In CNNs this is trivial as it is possible to just connect downwards to the previous layer in the same region.

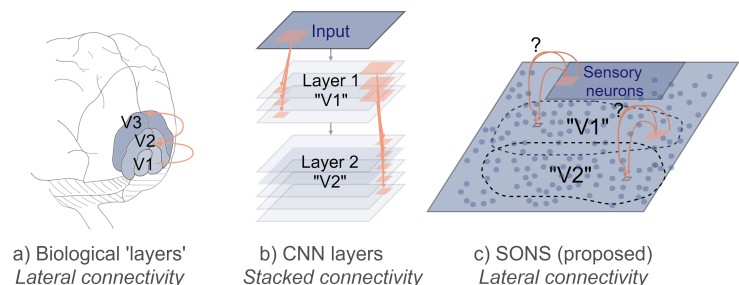

Figure 1: The visual cortex in the brain responsible for visual information processing has several regions which are laid out laterally next to each other (a) instead of stacked vertically like in standard CNNs (b). The proposed self-organizing neural sheet model (SONS) shown in (c) consists of just a single sheet of neurons which can self organize into local receptive fields and hierarchical layers. The main challenge in growing new connections laterally is to maintain the same spatial mapping from the input to other regions in the sheet. The marked V1 and V2 areas in our proposed model are not fixed, but emerge from self-organization rules.

However, this is no longer possible in a single sheet approach, as everything is laid out in just 2 dimensions. Achieving this type of lateral mapping offers greater flexibility to adapt to the input domain, as we are no longer constrained by predefined fixed layer and kernel sizes. Features are also encoded very differently in biological neural networks, forming topographic maps that tend to automatically cluster neurons that encode similar concepts closer together. This means neurons have a clear spatial importance that is mostly ignored in standard DNNs.

If we can derive self-organization principles that automatically form hierarchical layers we would no longer have to pre-design a neural architecture, or be constrained by fixed kernel sizes and the inherently predefined structure in convolution layers.

Our contributions in this paper can thus be summarized as follows:

- We propose a Self-Organizing Neural Sheet (SONS) model that consists of a single large pool of unstructured spiking neurons in 2D space. This model is capable of forming completely unsupervised its own network topology and localized topographic maps through a simple set of self-organization principles.

- We propose a dynamic rewiring method for spiking neural networks that can grow new synaptic connections and prune existing ones at run time. This allows us to optimize the wiring cost of the network by actively constraining the number of incoming and outgoing connections. Additionally, it provides us the ability to form large sparse synaptic weight matrices, making it possible to efficiently simulate tens of thousands of neurons and update millions of synapses in parallel.

- We derive a stochastic Integrate and Fire (Stochastic IAF) neuron model with synaptic weight normalization that results in a stable homeostatic neuron model capable of stochastically initiating synaptic growth.

- We show that the proposed stochastic integrate and fire model together with lateral excitation and inhibition and spike-timing-dependent plasticity (STDP) allows for the formation of logical topographic maps. These maps cluster neurons that encode similar features closer together.

- We show that the proposed SONS model consisting of just a single sheet of initially unstructured spiking neurons is capable of forming localized receptive fields and multiple hierarchical layers in a fully unsupervised manner.

## 2 Self-Organizing Neural Sheets

In order to achieve our desired objective of having a single sheet of spiking neurons self organize into logical topographic maps and different hierarchical layers we derive several self-organization principles. These involve a new type of stochastic integrate and fire spiking model with synaptic scaling to avoid exploding weights, lateral excitation and inhibition to form topographic maps, and dynamic rewiring rules to minimize overall wiring cost. The following subsections will describe each of these components in more detail, as well as the initial preprocessing steps.

### 2.1 Preprocessing

In order to properly take static images as input we first have to convert them into temporal spiking patterns (see figure 2). This is done by computing ON and OFF cell responses through center surround fields. These ON and OFF cells also exist in the brain [32], and respond naturally to blobs with either a dark or light center. These can be computed efficiently using standard Gaussian convolutions on the image. We can then bin the response of these cells (normalized between 0.0 and 1.0), and fire a spike when the bin index is equal to the current time step. For example if we have 10 bins and an ON cell has an output response of 1.0 it will fire at $t = 0$, while a response of 0.8 will fire at $t = 2$. This means higher intensity input will fire earlier. After this preprocessing step we can directly pass the spike responses of the ON and OFF cells to the neural sheet. These cells form the sensory input neurons. It is important to note that in contrast to traditional methods of connecting to the input in the previous layer, these sensory neurons are directly part of the single neural sheet.

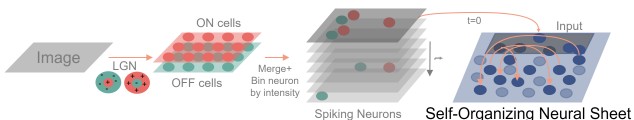

Figure 2: Conversion of a static image to temporal spikes. We first use center surround receptive fields to obtain ON/OFF cell output responses, which activate when either their center is light or dark. We then convert the real valued output of each cell into binary spikes by binning each value. Finally, the resulting input spikes at each time step become directly part of the entire neural sheet.

## 2.2 Stochastic Integrate and Fire neurons

In order to compute the output of each neuron in the neural sheet, we first calculate the change in membrane potential at each time step. The membrane potential is the value of the neuron before spiking. The change in membrane potential $du_i$ is simply the sum of the incoming spikes for neuron $i$. That is $du_i = \sum_{j=0}^{k} w_{ij} x_j$, where $w_{ij}$ the synaptic weight value, and $x_j$ a binary number indicating whether the incoming connected neuron $j$ spiked or not. We then get that $u_{t+1} = u_t + du_t$. This is similar to a standard forward pass in DNNs, except we have spikes as inputs, and we sum over time.

After obtaining the membrane potential we can apply a model responsible for generating spikes. Using a spiking based model allows us to only grow and update synapses of neurons that actually spiked, significantly reducing the overall computational requirements. We also desire a spiking neuron model that has some inherent stochastic behaviour in order to properly grow new synaptic connections from scratch. To this end we derive a stochastic leaky integrate and fire model (Stochastic IAF). The probability of this stochastic IAF firing scales proportionally to the membrane potential $u$. This means the higher the summated input, the more likely it is the neuron fires. The probability of the stochastic IAF neuron firing is as follows:

$$P(spike|u_t) = u_t^a + b \tag{1}$$

Where $a$ indicates a power parameter that suppresses low membrane potentials $u_t$ further (in our case $a$=5). This reduces the overall firing probability, as $u_t$ is always between 0 and 1 due to synaptic scaling. $b$ is an inherent bias, which gives a small probability of spiking to occur even when the membrane potential is zero (in our case $b$=0.0001). This essentially means that the higher the membrane potential is, the more likely it will fire. This was found to work well in practice, and results in spiking behaviour that is dependent on the actual membrane potential $u_t$, but also has sufficient stochastic behaviour to initiate synaptic growth. After spiking we reset the membrane potential to $-1.0$, which causes a hard refractory period. This makes it impossible for the neuron to fire again until it is reset back to 0, which happens when a new image is presented.

## 2.3 Spike-Timing-Dependent Plasticity (STDP)

Synaptic connection weights in the SONS model are modified through the standard STDP function [21, 22, 23], which takes into account both the pre and post synaptic spike timing. It increases the strength of the synapse if the post-synaptic neuron fires after the pre-synaptic neuron, and decreases it if the post-synaptic neuron fires before the post-synaptic neuron. This means causal correlations between neurons will strengthen the synaptic connection, making the neuron more likely to fire to similar input patterns in the future. The update equation to the weight matrix is as follows:

$$dW(t) = \begin{cases} Ae^{(\frac{-t}{\tau})}, & \text{if } t > 0 \\ -Ae^{(\frac{t}{\tau})}, & \text{if } t < 0 \end{cases} \tag{2}$$

where $A$ and $\tau$ are parameters that determine the shape of the STDP function. In all our experiments $A$ and $\tau$ were simply set to 1.0. $t$ is the relative timing difference between the pre and post synaptic spike and $dW$ is the resulting change in the synaptic weight value .

For updating the weights we get $W_{t+1} = W_t + \eta \frac{dW}{d(p,q)}$ where $\eta$ is the learning rate and $d(p,q)$ is the distance between the neurons of a given synapse. The weight update is thus scaled by the synaptic length, meaning longer connections take more time to achieve high values, making it more likely they get pruned. In our experiments $\eta$ was set to 0.1.

## 2.4  Synaptic scaling

To achieve a type of homeostatic balance in the neurons, synaptic weight values are rescaled to have a fixed norm value. This is sometimes referred to as synaptic scaling and it also occurs in biological neural networks [33]. This ensures that the weights do not grow indefinitely causing an explosion in the membrane potential and uncontrolled spiking behaviour. Rescaling is simply done by dividing all the incoming weights of each neuron by the L2 norm after each STDP update to the weights. This is similar to weight normalization [34] in standard neural networks, which also helps with regularization in those cases. The weight are thus rescaled as follows:

$$w_{ij} = \frac{w_{ij}}{||\boldsymbol{w_i}||_2} \tag{3}$$

Where $||\boldsymbol{w_i}||_2$ is the L2 norm over all incoming synaptic weights $j$ of neuron $i$. This also allows us to directly use a probabilistic spiking model, as the norm and resulting membrane potential never exceeds 1. Since we have a stochastic spiking model we can still spike when the membrane potential reaches close to 1, so we do not require all inputs to be active in order to reach the threshold.

## 2.5  Lateral excitation and inhibition

To overcome the limitation in the classical SOM model, where only a single unit is allowed to be active, we instead use lateral excitation and inhibition. This allows us to learn multiple topographic maps within the sheet, and form proper layers and localized receptive fields. Lateral excitation and inhibition can be efficiently implemented through a difference of gaussian kernel (DoG), which consists of a positive Gaussian kernel with a small standard deviation subtracted by another Gaussian kernel with a larger standard deviation.

This lateral excitation and inhibition method forces close neighbours of spiked neurons to activate, causing them to spike as well. This results in a type of spike wave around the initially activated neuron. After the neighbours spike, they will also undergo a STDP update on their synaptic connections and thus form similar connections to the same input. Neurons close together will therefore automatically learn similar features. The larger overall lateral inhibition ensures only a localized subset of neurons activate, which results in a type of localized winner take all strategy. More details of the exact equations and parameters used are given in the appendix.

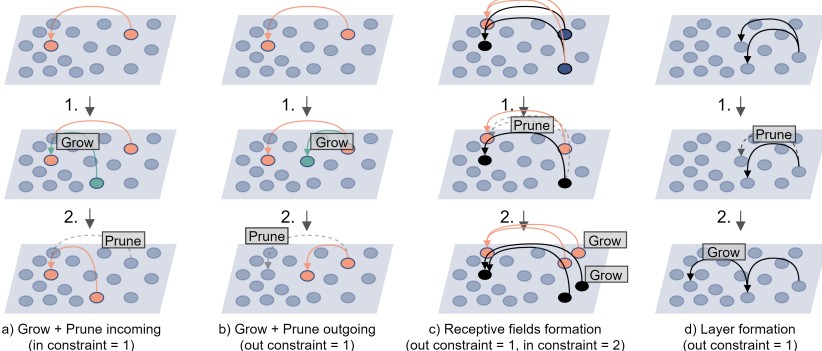

Figure 3: Examples of how neurons can self-organize by minimizing the overall wiring cost through dynamically growing and pruning existing connections under connection constraints. In case of (a) and (b), When connection constraints are violated through new growth (a.1, b.1), synapses are pruned by their highest wiring cost (a.2, b.2), keeping only the best newly grown connections found. Receptive fields can also form in this manner (c). When outgoing constraints are violated, connections are pruned (c.1), and new connections can be grown from unused neurons (c.2). Minimizing the wiring cost under constraints thus causes neurons to automatically distribute their responsibility. Layer formation similarly can emerge (d). When outgoing constraints are violated connections are pruned (d.1), and different connections can grow (d.2). Since the number of allowed outgoing connections is limited new connections can only be made to neurons higher in the hierarchy.

## 2.6 Dynamic rewiring and synaptic competition

Our SONS model also has a dynamic rewiring component, which dynamically grows and prunes synaptic connections. This allows us to significantly scale up the network, as we no longer need fully connected weight matrices. Additionally, hard constraints on the number of allowed incoming and outgoing connections causes local competition between neurons, where weaker connections are pruned, while stronger ones are kept. If we prune connections based on the strength and length of the synapse, it automatically causes the network to indirectly minimize the overall wiring cost. This process of minimizing the wiring cost inherently results in the emergence of a type of hierarchy, since neurons only have a limited number of allowed outgoing connections. It also causes neurons to automatically distribute their receptive fields, as they have to compete to connect to each input sensory neuron (see figure 3).

The growth process could be seen as a type of Monte-Carlo sampling [35], where every $m$ time steps we sample new possible connections. Each neuron that spiked grows new connections to other neurons that also recently spiked. This lets us grow connections to new neurons that already have a high potential to correlate. We also only grow some minimum distance from the neuron and only grow in a single direction, this ensures we do not connect purely to local neurons that are already activated through lateral excitation or create complex feedback loops. A higher minimum distance essentially enables larger local topographic maps to emerge. In our experiments we set the minimum distance to 20, within a neural sheet of 100x200. This number allows for around 10 layers to emerge.

Each time we grow new connections, we first prune existing ones to make space (if we violate a constraint on the number of allowed connections). Pruning is done by sorting the incoming and outgoing connections of each neuron $i$ by their synaptic weight value. We then temporarily freeze the top-k weights with the highest magnitudes $|w_{ij}|$, and remove all other connections to free up space. In our case $k$ was set to $64$ for the incoming and outgoing connections, which allows for 8x8 kernels to form. This also makes overlapping receptive fields and local synaptic competition possible. For the total capacity we allow $3k$ synapses, letting us grow $2k$ synapses per step.

## 3 Method

We evaluate our method with several experiments. First we generate topographic maps by simulating a sheet of 32x32 spiking neurons. We compare results of our proposed Self-Organizing Neural Sheet (SONS) with standard spiking neural networks (SNNs) and spiking self-organizing maps (SOM). As input we take 7x7 patches from the standard MNIST dataset [36] and convert them to temporal spike patterns as was described in figure 2. Although the dataset is relatively simple, it can still serve as a good baseline for the evaluation of spike-based methods, and allows for easier analyses of the SONS model behaviour. Each time a new image is presented all membrane potentials are reset back to zero to ensure a clean slate. Patches are presented for 20 time steps, and in total the sheet is simulated for 200k timesteps until convergence (applying the self-organization rules at each step). We then evaluate our method on the full images. In this case we use a large sheet of 100x200 spiking neurons (20k neurons). Images are presented for 60 time steps, and the entire sheet is simulated for 3600k time steps until convergence. Existing methods fail in this case as they require fully connected weight matrices. However, we do perform ablation studies in order to study the effect of each component. The implementation was mainly done in Pytorch [37], together with a custom cuda-accelerated data structure for efficiently growing and pruning connections. All experiments were run on a single Titan V and 12-core i7 CPU. Please see the appendix for full implementation details.

**Evaluation metric**    To evaluate the quality of the topographic maps and resulting synaptic connections we can calculate both the synaptic strengths of the top-k strongest connections and the length of each of those connections. Ideally the strongest connections go up in strength while reducing in length. High strong local connectivity indicates that nearby neurons are highly correlated, which is the desired property in good topographic maps. This allows us to derive the following desired loss metric which also indicates the overall wiring cost of the network.

$$L = \sum_{i=0}^{n} \sum_{j=0}^{k} \frac{d(p_i, q_j)}{w(p_i, q_j)} \tag{4}$$

Where $n$ is the number of neurons, and $k$ the number of synapses per neuron. $p$ and $q$ are points indicating the $xy$ positions of each neuron. So $w(p_i, q_j)$ is the synaptic weight between neuron $i$ and $j$ as learned through STDP given in eq. 2 and normalized through eq. 3, and $d(p_i, q_i)$ is simply the euclidean distance between neuron $i$ and $j$.

## 4  Results

To keep comparisons fair, we use the same underlying implementation and experimental setup for all our experiments. The main difference of SONS to SNNs is that SONS have lateral excitation and inhibition together with dynamic growth and pruning of synaptic connections. SOMs do have lateral excitation and inhibition but no dynamic connections, they can thus form topographic maps, but are unable to form receptive fields or hierarchical layers. SNNs and SOMs also require fully connected weight matrices, and thus scale poorly to a large number of neurons.

**Emergence of topographic maps**    The resulting learned topographic maps can be found in Figure 4. Note that the sensory neurons do not have weights or connections and therefor do not encode any features and are not visualized. It can be seen that SNNs learn proper edge filters as expected, but do not self-organize into logical clusters. SOMs and SONS do form more organized representations, where similar edges cluster closer together. Training progress can also be found in figure 5 and the final performance is given in Table 1. This shows that our proposed SONS also further minimize wiring cost as opposed to SOMs (p<.001, two-tailed t-test, calculated over 5 runs), meaning features are better organized such that it minimizes the overall wiring length toward the input. It's also important to note that although our model and SOMs obtain similar results, SOMs do not scale to larger sheet sizes, and only work within this single-layer experiment (in contrast to SONS).

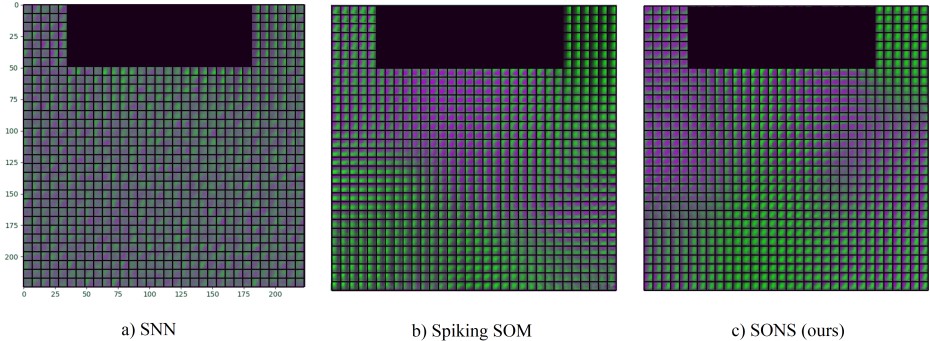

a) SNN                          b) Spiking SOM                    c) SONS (ours)

Figure 4: Comparison between the learned topographic maps of different models. Visualized is the learned features of each spiking neuron in a 32x32 grid. Each feature is the same size as the input (7x7), where green indicates a connection to an OFF cell, and purple to an ON cell. The black region marks the sensory input neurons, which have no connectivity. Compared to standard SNNs, neurons in our SONS model can be seen to form more logical clusters, clustering learned edges of similar orientations closer together. Spiking SOMs perform similarly in this setting, but they only work on single layers, are not scalable to larger sheets, and they do not minimize the overall wiring cost.

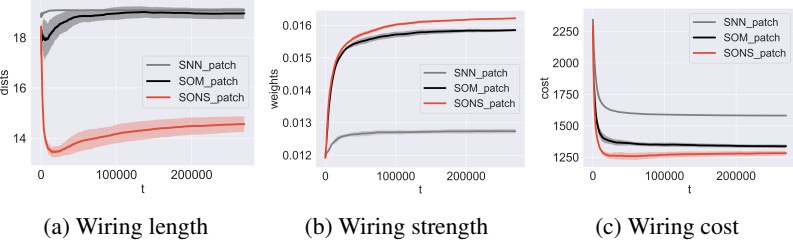

(a) Wiring length                (b) Wiring strength               (c) Wiring cost

Figure 5: Training progress in terms of mean synaptic connection length ($d(p, q)$), mean synaptic strength ($w(p, q)$) and total resulting wiring cost ($\frac{d(p,q)}{w(p,q)}$). The shaded area is the standard deviation.

| | Patch-7x7 (sheet 32x32) | | | Image-25x25 (sheet 100x200) | | |
|---|---|---|---|---|---|---|
| | *w(p,q)* | *d(p,q)* | *Cost L* | *w(p,q)* | *d(p,q)* | *Cost L* |
| SNN | 0.0127 (±8e-5) | 19.08 (±0.03) | 1581 (±5.61) | N/A | N/A | OOM |
| Spiking SOM | 0.0158 (±4e-5) | 18.96 (±0.22) | 1337 (±14.85) | N/A | N/A | OOM |
| **SONS (ours)** | **0.0162** (±3e-5) | **14.56** (±0.32) | **1282** (±19.02) | **0.0131** (±7e-4) | **27.78** (±5.01) | **2076** (±76) |

Table 1: Comparison between different models on both small image patches using a grid of 32x32 neurons, and full images using a grid of 100x200 neurons (20k neurons). SNN and Spiking SOMs require fully connected weight matrices and run out of memory in larger sheets (OOM). Our proposed SONS model on the other hand can still be simulated even for such larger sheets.

**Emergence of receptive fields and layers** Figure 6 illustrates the full image experiment using our SONS model. The total simulated sheet consists of 20k neurons organized into a 100x200 grid. Receptive fields and layers can be seen to emerge within the single sheet of spiking neurons. The network is initially completely random and unstructured, but it can be seen that the spiking neurons properly self-organize to be responsible for different parts of the sheet after training. Neurons in the sheet properly connect locally to nearby neurons, minimizing the overall wiring length. Neurons grow localized receptive fields and properly maintain the spatial mapping across the sheet. Topographic maps of edge filters also emerge. These are smaller localized edge filters that logically cluster together by their orientations. They also connect to the same part of the visual field.

A visualization of the spiking behaviour within this sheet can be found in figure 7. Spikes can be seen to propagate from the input across multiple logical hierarchical 'layers'. These layers emerged completely independently and were not predefined. By calculating the average spike response to each digit we can also observe higher order neurons that properly cluster similar digits together, which means topographic maps also form at higher abstraction levels.

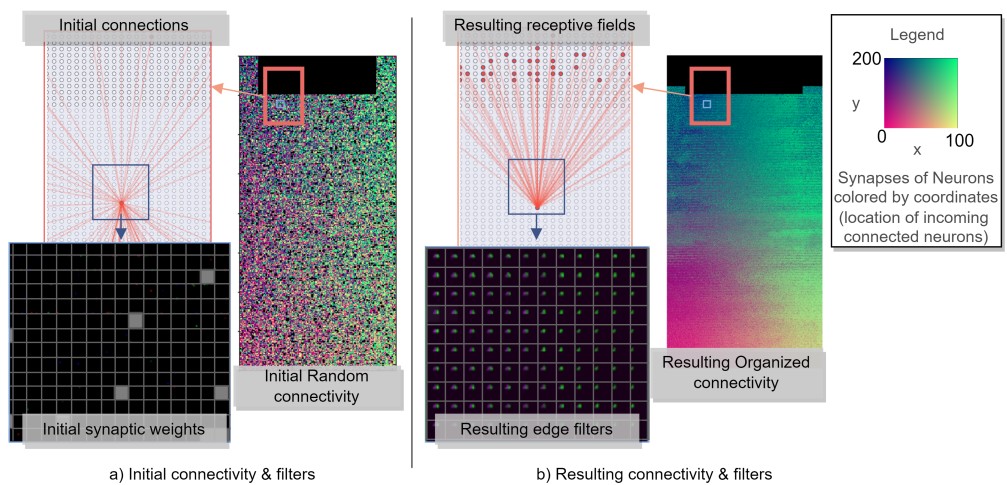

Figure 6: Results showing how the self-organizing neural sheet forms localized receptive fields and localized topographic maps. Colors indicate the position of each synapse in a 100x200 spiking neuron sheet (see legend). The positions of the strongest 64 synapses of each neuron are visualized by reshaping the connections into a 8x8 patch and coloring them according to this color mapping (as seen in resulting organized connectivity). For example, purple colors indicate that the synapses of the neuron in that position are mostly connected to the bottom left of the neural sheet. The actual weights of each synapse are also visualized by zooming in on several neurons close to the sensory neurons (resulting edge filters), which show ON/OFF cell contrasted edge type filters of different orientations. The connectivity of a single neuron is also highlighted (resulting receptive fields), which shows it formed localized receptive fields that maintain a proper spatial mapping to the input. The black area at the top indicates the input sensory neurons, which get overwritten with the actual spiking pattern of the presented stimuli image (it has no incoming synapses). Initially the sheet is completely randomly connected with no meaningful receptive fields or edge filters (a). Through our derived self-organization rules the sheet converges to an organized connectivity pattern, where synapses from each neuron connect to the same region and form meaningful edge-like feature detectors (b).

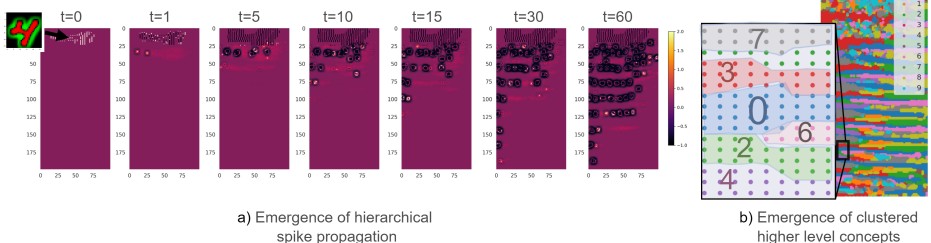

a) Emergence of hierarchical spike propagation

b) Emergence of clustered higher level concepts

Figure 7: Example of how spikes propagate throughout the neural sheet, where darker regions indicate which neurons have previously spiked at each time step (a). This spike propagation is hierarchical, where higher-order neurons only fire after those closer to the input have fired. This is a direct result of the localized organized connectivity that emerges from our self-organization rules presented in figure 6. It can also be observed that presenting a single digit activates less and less neurons farther from the input, which can be explained by the emergence of higher level abstract concepts. Farther from the input, neurons self-organize into logical clusters, clustering similar digits closer together (b). In this case the different colors indicate the label of the input to which each neuron at that position spikes most often. It can be seen that at the top of the sheet (closest to the sensory input neurons) the neuron activations are mostly random. However, at higher order neurons farther from the input logical clusters quickly emerge, where nearby neurons tend to fire to the same digit.

**Ablation studies** Experiments were also performed to evaluate the importance of each of the components of the self organization rules. Removing the STDP update naturally results in a static network where no learning occurs and all weights stay the same. Removing synaptic weight normalization destroys the homeostasis of the spiking neurons. Without some regularization or balance on the synaptic weights the weights just grow indefinitely, causing all neurons to always spike. Removing the dynamic growth and pruning of synaptic connections results in a statically wired network. It is still possible to learn very sparse individual features, however no receptive fields or meaningful layers can form and the network is just randomly wired. Also, without lateral excitation and inhibition no topographic maps form, similar to the case of using standard SNNs in figure 4. Additional details related to these experiments can be found in the appendix.

## 5  Discussion

We showed that our derived self-organization rules are capable of producing proper hierarchical layers and topographic maps. The biggest limiting factor of the results obtained is that our model requires extreme scalability. We were able to simulate 20 thousand neurons, but the visual cortex has over 100 million neurons, and the entire cortex has more than 20 billion [38]. This means the size of the topographic maps we obtain is fairly limited. However, this type of single sheet model is inherently extremely scalable, as we can simply increase the entire neural sheet size. Further experiments would have to be done to investigate the extend to which we can scale, and to confirm what type of self-organized behaviour we might obtain in extremely large networks. Additionally, It would be interesting to explore alternatives to the implemented lateral inhibition to be more biologically plausible, allowing for more unique lateral connectivity in different regions of the neural sheet.

There are also several possible future research directions to further improve the proposed self-organizing neural sheets. Since the network consists purely of simple spiking neurons, it would be interesting to apply it to neuromorphic chips or more specialized hardware [39], which could allow for much greater scalability and reduce the overall environmental impact of training extremely large models. Another direction is to allow for spatio-temporal maps to form that can learn topographic maps of temporal sequences. It would then be possible to apply it to different input modalities (e.g. audio waves). Better visualization methods would also be desirable, to gain a better understanding of the behaviour of each neuron in the sheet and overall neural spiking dynamics [40].

This paper presents a first step towards a different type of neural network that consists just of a large single sheet of self-organizing spiking neurons. This neuronal sheet does not require any manually predefined neural architecture of layers and connections, but it can instead dynamically rewire and form new connections and hierarchical layers as needed.

## Acknowledgments and Disclosure of Funding

This work was supported by Institute of Information & communications Technology Planning & Evaluation (IITP) grants funded by the Korea government(MSIT) (No. 2019-0-00079, Artificial Intelligence Graduate School Program, Korea University, No. 2019-0-01371, Development of Brain-inspired AI with Human-like Intelligence, No. 2021-0-02068, Artificial Intelligence Innovation Hub, and No. 2022-0-00984, Development of Artificial Intelligence Technology for Personalized Plug-and-Play Explanation and Verification of Explanation).

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
