# OpenReview forum: "Emergence of Hierarchical Layers in a Single Sheet of Self-Organizing Spiking Neurons"
_NeurIPS.cc/2022/Conference — NeurIPS 2022 Accept_

### Official Review · Reviewer_BJ27 · 2022-07-10

**Rating:** 6
**Confidence:** 4
**Soundness:** 3 good
**Presentation:** 3 good
**Contribution:** 3 good

**Summary:**

This paper proposes an interesting model to form visual cortex-like hierarchical layer structures in a self-organization manner. Different from traditional neural networks that ignore the placement of neurons, this study accounts for the location of the neurons in learning with the Self-Organizing Neural Sheet (SONS) model, inspired by the mechanisms of LGN cells (Figures 2 and 3). Experiments with MNIST dataset demonstrate that the SONS model can learn visual cortex-like topographic maps (Figure 4) at a lower cost (wiring cost, Figure 5) compared with SOMs. And the learned structure presents clustered higher-level concepts (Figure 7).

**Questions:**

1.	As shown in Figure 4, using spiking SOM can learn similar patterns, only that SONS has a lower wiring cost. Please explain why the wiring cost is a proper metric for evaluation. And what is the advantage of the proposed SONS model?
2.	In Figure 7b, what does the lower level (near to the input) features like?


**Strengths And Weaknesses:**

Strength:

The topic is interesting and the idea of taking the placement of neurons into account is of significance. Especially, results demonstrate that the model not only learns visual cortex-like topographic maps but also the layer-wise patterns from low-level patches to the concept of objects is valuable and inspiring. This type of thinking is beneficial for novel network design for a wide area.

Weakness:

The MINIST dataset is too simple to reflect rich information in the network learning process. More image datasets with natural pictures should be evaluated to better support the claims.

---

> ### Author Response · Authors · 2022-08-02
> **Response to reviewer BJ27**
>
> Thank you for the generally positive comments and feedback. We have given a detailed point-to-point response to your comments below. Hopefully these will resolve most of your remaining concerns, and that they can be taken into consideration when deciding the final review score of the paper.
>
> ---
>
> “The topic is interesting and the idea of taking the placement of neurons into account is of significance. Especially, results demonstrate that the model not only learns visual cortex-like topographic maps but also the layer-wise patterns from low-level patches to the concept of objects is valuable and inspiring. This type of thinking is beneficial for novel network design for a wide area.”
> - We also believe our proposed SONS model can indeed have larger implications for a wider area in different domains and more complex datasets, and that we provided an initial step in such a direction. Future directions could include for example different modalities (sound, sequence data etc.), and a further exploration of how the connectivity relates to the actual coretex in biological neural networks.
>
> “The MINIST dataset is too simple to reflect rich information in the network learning process. More image datasets with natural pictures should be evaluated to better support the claims.”
> - The paper was updated to explain this a bit further (line 215). Our main focus was on describing and analyzing the proposed SONS model, since we wanted to create an initial foundation for a fundamentally different type of neural network. A smaller and relatively simpler dataset made it easier to analyze and understand the neural sheets behavior. The proposed SONS model hopefully creates an initial foundation for future work that could explore more diverse datasets and different domains.
>
> “As shown in Figure 4, using spiking SOM can learn similar patterns, only that SONS has a lower wiring cost. Please explain why the wiring cost is a proper metric for evaluation. And what is the advantage of the proposed SONS model?”
> - For biological brains the wiring cost is likely minimized due to space limitations, less wires means less space, and also less computation and energy. This is touched upon in more detail in section 2.6. A low wiring cost means fewer strong connections, placing constraints on the number of strong incoming and outgoing synapses. It also results in shorter localized connections, creating a more inherent hierarchy. This is why we used it as our evaluation metric. In our SONS model, the advantage is similar. A lower wiring cost means less synapses are required (reducing memory and compute time), and connections are shorter, meaning connectivity is more localized (allowing the inherent formation of hierarchies).
>
> “In Figure 7b, what does the lower level (near to the input) features like?”
> - Features in the lower level are visualized more clearly in Figure 6b. The zoomed in visualizations of the neurons in those regions show that they form edge-type like filters, so they respond more strongly to input patterns that look like edges of similar orientation.

---

> > ### Comment · Reviewer_BJ27 · 2022-08-09
> > **Response to the authors**
> >
> > I thank the authors for the response. I agree that the MNIST dataset can reflect the performance more easily, while I still think it is quite limited since the images are handwritings rather than natural images. Overall, I appreciate the idea of the paper, and the presentation is also clear and technically sound. I will keep my scores at the point, but further discussion is welcome.

---

### Official Review · Reviewer_htWu · 2022-07-10

**Rating:** 8
**Confidence:** 4
**Soundness:** 4 excellent
**Presentation:** 4 excellent
**Contribution:** 3 good

**Summary:**

The paper presents an unsupervised method that allows the formation of local receptive fields and hierarchical layers in an initially unstructured pool of spiking neurons.

**Questions:**

None

**Limitations:**

The authors adequately addressed the limitations of their work.

**Strengths And Weaknesses:**

The method is novel and is based on purely local learning rules with lateral inhibition/excitation. This makes it especially valuable for those interested in biologically plausible approaches. The presented method performs similar to self-organizing map but shows better scaling to larger populations of neurons. This, together with the biological plausibility of the method, makes the presented work a strong contribution.

The paper is well positioned using relevant benchmark problems and comparisons to alternatives. It would however be nice to see an additional experiment other than MNSIT.

The review of previous work is very strong. Throughout the paper the authors show a strong familiarity with the relevant literature. This is a strong foundation that makes everything else very convincing. Experimental work is carefully done.

Additional comments:
-L109: "we fist we have"

---

> ### Author Response · Authors · 2022-08-02
> **Response to reviewer htWu**
>
> Thank you for your positive comments and feedback, we also believe our proposed SONS model establishes a strong foundation for future works that can hopefully further expand and fully explore additional applications of the model.
>
> ---
>
> “The paper is well positioned using relevant benchmark problems and comparisons to alternatives. It would however be nice to see an additional experiment other than MNSIT.”
> - The method section was updated to clarify this a bit further (line 215). We chose to focus on first establishing an initial foundation for our proposed SONS model. A relatively simpler datasets allowed us to better analyze and understand the neural sheets behavior, which we felt was more important. It also allowed us to still stay within the given page limits, and prevented the scope of the paper from getting too big. Hopefully we established a good foundation for future works that can expand and apply our model to more complex datasets and other domains as well.
>
> “Additional comments: -L109: "we fist we have"”
> - Thank you, we fixed this.

---

> > ### Comment · Reviewer_htWu · 2022-08-09
> > **Response to authors**
> >
> > I thank the authors for their response. I originally thought that this paper should be accepted. Implementing changes suggested by the other reviewers will further improve the manuscript. I stand by my score.

---

### Official Review · Reviewer_zRzp · 2022-07-11

**Rating:** 5
**Confidence:** 5
**Soundness:** 3 good
**Presentation:** 3 good
**Contribution:** 2 fair

**Summary:**

The paper is based on an original idea that consists in considering a neural network organized on a simple neural layer, and to try to obtain the emergence of a hierarchical organization in this layer like the one observed in classical networks like "LeNet". The paper is properly organized and describes the model from the transformation of an image into a spike pattern  then to the stochastic neuron model, a learning rule of type STDP and the construction of lateral interactions both excitatory and inhibitory. The method is then described on a model with a limited number of spiking neurons, and then shows results for the emergence of a topographic map. This model is compared with a classical self-organized map and shows superior results in this model. This model finally concludes on the emergence of a hierarchy in the map, which is illustrated in figure 7 by the propagation of spikes on the neural map.




**Questions:**


I have several questions in relation to this paper.
First, have you looked at different settings of the topography constraints in the neural layer to see the emergence of different structures? There must be an optimal trade-off for a given task in order to achieve a hierarchy.
Second, in section 2.4 you use a synaptic scaling rule that is similar to a homeostatic rule. Have you looked to see if you actually get neuron discharge frequencies that stabilize in the neuronal layer?
Third, it would be interesting to study the emergence of waves of activity in your neuronal layer. This kind of pattern in spontaneous activity seems to be essential in the emergence of the structure of a hierarchical type network and could be an interesting extension to your model


**Limitations:**


The authors did not specify the limitations and potentially negative societal impacts of this paper.


**Strengths And Weaknesses:**


The main strength of this paper is the originality of the idea of organizing the whole neural network in the same way as the organization of the cortex along a surface. However, this paper is relatively limited by the small size of the network used and also by the simplicity of the task that is performed. The emergence as announced in figure 7 seems too preliminary to be convincing. One can also wonder what is the usefulness of using a spiking neural network in this framework, especially since neural models are based on stochastic firing patterns that potentially destroy the temporal structure of the sensory input.

---

> ### Author Response · Authors · 2022-08-02
> **Response to reviewer zRzp**
>
> Thank you for your generally positive comments and feedback.
>
> - We have updated the paper to further explain the importance of spiking neurons in our model with regards to synaptic growth and computational efficiency, which hopefully further clarifies the use of spiking neurons (line 128).
>
> There is always a trade-off between the page-limit and exploring all the aspects of the proposed SONS model, so some of the points would be difficult to address without the paper quickly growing out of scope. We do believe we provided an initial first step to a fundamentally new type of neural network, that can hopefully serve as a foundation for future works that can further expand upon our model to be applied to more complex datasets, and other modalities (sound, sequences etc.).
>
> Below we have also provided a point-to-point response to some of your questions and concerns raised.
>
> We hope these sufficiently clarified some of the aspects of the paper, and that they can be taken into account when deciding the final review score.
>
> -------
>
> “The main strength of this paper is the originality of the idea of organizing the whole neural network in the same way as the organization of the cortex along a surface. However, this paper is relatively limited by the small size of the network used and also by the simplicity of the task that is performed. “
> - We updated the method section to clarify this a bit further (line 215). Since our main focus was to establish a fundamentally different type of neural networks, we chose to keep the task itself relatively simple. This allowed us to focus mostly on describing and analyzing the proposed SONS mode, and it made it easier to understand the neural sheets behavior. We believe the proposed model and task shows the potential of the approach, and that we provided an initial first step and a foundation for future work in using SONS on more complex datasets, and other modalities as well.
>
> “One can also wonder what is the usefulness of using a spiking neural network in this framework, ...”
> - Since the spiking behavior triggers the growth process of new synapses, it significantly reduces the computational requirements. With a spiking model we only have to compare against the neurons that actually spiked, instead of against all neurons. We updated the paper to clarify this further (line 128).
>
> “ First, have you looked at different settings of the topography constraints in the neural layer to see the emergence of different structures? There must be an optimal trade-off for a given task in order to achieve a hierarchy.”
> - We did perform ablation studies on some of the parameters and components, and the importance of each of them. These are also mentioned within the results section (line 268) and further details can also be found in the appendix (appendix table 1). The hierarchy is mostly governed by the constraints on the number of outgoing connections (described further in figure 3). If we had no such constraints, all neurons could simply connect to the sensory input neurons, and no hierarchy would form. Our chosen parameters were mostly picked according to logical requirements, but it is indeed possible more optimal parameter values exist within different settings, which would be interesting to explore further in future works.
>
> “Second, in section 2.4 you use a synaptic scaling rule that is similar to a homeostatic rule. Have you looked to see if you actually get neuron discharge frequencies that stabilize in the neuronal layer?”
> - Figure 7a gives a visualization of how spikes propagate across the sheet. This shows they do seem to stabilize, and are not constantly chaotically spiking. Constant spiking would prevent proper STDP and synaptic growth, and the homeostatic rule was found to be sufficient in our setup. It could however be interesting to investigate different homeostatic rules in future work, that do not rely on synaptic weight normalization, and to further investigate discharge frequencies depending on the input and whether they are stable under all conditions.
>
> “Third, it would be interesting to study the emergence of waves of activity in your neuronal layer. This kind of pattern in spontaneous activity seems to be essential in the emergence of the structure of a hierarchical type network and could be an interesting extension to your model”
> - We do observe some waves in figure 7a, that propagate hierarchically along the sheet from the input. These waves are naturally generated through the Lateral excitation and inhibition component, since they excite local neurons (some additional details can also be found in the appendix), and seem to be essential for the formation of topographic maps. It would indeed be an interesting future research direction to explore these waves further, and to relate them back to biological neural networks, but this would not have fit well within the scope of our paper.

---

### Official Review · Reviewer_e8kL · 2022-07-14

**Rating:** 7
**Confidence:** 4
**Soundness:** 2 fair
**Presentation:** 3 good
**Contribution:** 3 good

**Summary:**

Although the cortex can be approximated as a 2-D sheet, different regions of the sheet have been identified as being connected together hierarchically, such as V1 > V2 > V4 ...  The authors develop a stochastically spiking sheet model with lateral connectivity and simple rewiring rules that forms a biologically analogous hierarchical structure--that are mimicked by deep ANNs--in an unsupervised manner.

**Questions:**

1- Eq 1: How robust are the finding to different a values (not equal to 5)?

2 - Does this require "lateral inhibition", i.e., longer inhibition than excitation, as described in the model?  Such connectivity is not supported by the evidence, and so if it does, this assumption should be listed as a limitation of the current model

3 - L185 describes a hard constraint on the incoming and outgoing connection numbers.  Can this requirement be softened to a more bioplausible one?  E.g., perhaps a range of connections would suffice, and be more realistic

4 - L203: Similarly, might this constraint be softened?

5 - Fig 7b: it appears that around 7-9 layers have formed; is this a function of the distance connectivity of the network + the number of inputs?  If so--especially the latter--the authors should address this aspect explicitly, as an input-dependent number of layers (/cortical regions) would amount to a testable prediction of their model in cortex

6 - How does this model interface with the known anatomical cortical laminae connectivities?  E.g., higher regions such as V4 are known to feed back to lower regions at the L1 apical dendrites that connect to L2/3 and L5 neurons.  Similarly, V1 connects to V2 and higher at L4 basal dendrites. Is such anatomical selectivity not enough to account for the regional distinction that arises in cortex?


**Limitations:**

Yes, but some limitations (see Strengths and Weaknesses and Questions above) may not have been addressed yet

**Strengths And Weaknesses:**

**Originality:**

 I am unaware of other work that describes how networks of neurons might form hierarchical structures similar to those in the brain from simple 2-D non-hierarchical initial conditions.

**Quality:**

The project has a clearly defined motivating question, methodology to address the question, and analyses that help to address the research question. However, no statistical analyses are done.  Fig 5 in particular appears to only illustrate single runs.  In 5c especially, it appears that the SONS patch might result in a larger wiring cost than the SOM patch.  What happens if these are run longer?  These graphs should be based off of multiple runs of the network, and standard deviations or errors should indicate the range of metrics that result.  Moreover, statistical analyses should be implemented to reveal, after the networks have converged, what resultant differences among the networks are statistically significant.  Minor note: there are a few places, such as L262, where the forward quote is backwards (presumably because of Latex's conventions). Please see the question section as well

**Clarity:**
The authors describe their approach and results clearly. Note: Fig. 4 is difficult to view, in part because of the size, in part because of the pallette chosen (I am red-green color blind, and find these particular red, green choices to be difficult to discriminate).  Additionally, it would be helpful to provide zoomed-in images of regions of interest that are pointed out in the caption and text.  For Fig 7a, please incorporate a colorbar legend.

**Significance:**

At the interface of deep learning and neuroscience, different cortical regions, such as V1/V2/V4/etc., are generally treated as layers in deep network models of such systems as the visual system with increasing amounts of success.  The authors here have asked an important question as to how such hierarchies arise in cortex, given that they structurally can be modeled as lying on a 2-D sheet, a question with interesting implications with respect to the design feedforward ANNs, as the authors point out.  Their analyses seem to provide a first step towards answering this question.  However, the authors to not address the known anatomical connectivity (feedback to apical L1 dendrites, feedforward to basal L4 dendrites) (Q6, below).  The significance of the present work would be made more apparent if the authors address why such known connectivity is insufficient to explain the segregation of cortical regions, and how their model is or might be made to be consistent with this known cortical microcircuit anatomical structure.

---

> ### Author Response · Authors · 2022-08-02
> **Response to reviewer e8kL**
>
> Thank you for the helpful comments and feedback. We were able to make several improvements as a result of your questions and comments. Mainly we modified the following:
>
> - Updated figure 5 to include mean and standard deviation of multiple seeds, and provided significance scores between the SONS and SOM model (p<.001).
> - Changed the color palette of some of the figures to be more colorblind friendly.
> - Expanded the discussion section slightly related to lateral inhibition and biological plausibility.
>
> We hope these comments and improvements can be taken into account when deciding on the final score for the paper. Please find below a more detailed point-to-point response as well.
>
> -----------
>
> “In 5c especially, it appears that the SONS patch might result in a larger wiring cost than the SOM patch. What happens if these are run longer?”
> - We did do some experiments where the model runs for much longer, and the average weight values, distances, and wiring cost quickly flatten out and do not improve much. We increased the limit of the plot such that this is a bit more clear (more obvious in the wiring strength and distance).
>
> “These graphs should be based off of multiple runs of the network, and standard deviations or errors should indicate the range of metrics that result. ... “
> - We have updated figure 5 as suggested to include a visualization of the mean and standard deviations over multiple runs (5 seeds), and calculated p-value using standard two-tailed t-test between the SOM and SONS model (p<.001) (also updated in line 250, and the base statistics can be found in table 1).
>
> “Note: Fig. 4 is difficult to view, in part because of the size, in part because of the pallette chosen (I am red-green color blind, and find these particular red, green choices to be difficult to discriminate). ...”
> - We have modified the color palette to be more colorblind friendly, and also slightly increased the base figure size.  Zoomed in versions of each of the plots were also added in the appendix. We also added a colorbar legend to figure 7a as suggested. We hope these improve the clarity of the figures.
>
> “1- Eq 1: How robust are the finding to different a values (not equal to 5)?”
> - The results are relatively robust to small changes in this value, but finding the optimal would likely require a more extensive grid search. More details on the exact spike probability curve are given in the appendix.
>
> “2 - Does this require "lateral inhibition", i.e., longer inhibition than excitation, as described in the model? ...”
> - The lateral inhibition is indeed required to form proper topographic maps. This type of connectivity causes wave patterns to emerge that activate nearby neurons, which ensures they will also learn similar features. More details on this are also given in the appendix. We have expanded the discussion section slightly to mention potential future research directions that explore this lateral connectivity implementation further to make it more biologically plausible (line 284).
>
> “3 - L185 describes a hard constraint on the incoming and outgoing connection numbers. Can this requirement be softened to a more bioplausible one? ...”
> - The hard constraint was mostly made for computational efficiency and easy pruning, since it allows a fixed size matrix for all neurons. However, it could potentially be possible to apply a soft constraint within this hard constraint, that looks purely at the synaptic weights.
>
> “4 - L203: Similarly, might this constraint be softened?”
> - This could also be potentially softened within the hard constraint.
>
> “5 - Fig 7b: it appears that around 7-9 layers have formed; is this a function of the distance connectivity of the network + the number of inputs? ...”
> - It does not necessarily directly depend on the number of inputs, however a larger input would also require longer connections to properly connect to each input sensory neuron. We added a slight clarification to section 2.6 to more explicitly link the relation between layers and the distance as suggested (line 202).  It might be possible to make this a more variable or learnable parameter in the future, but we did not explore that much further.
>
> “6 - How does this model interface with the known anatomical cortical laminae connectivities? ...”
> - The main focus of the paper was to establish a new type of neural network model that does not require a pre-defined hierarchical architecture (as is common in almost all deep learning based models), and some decisions were sometimes made for computational efficiency rather than full biological accuracy.  We do believe we established a new type of approach that could serve as a foundation for future papers that could further explore more accurately matching biological behavior (e.g. more complex spiking models, soft constraints, feedback loops, interaction between layers etc.). This was outside the scope of our paper, but it would definitely be interesting to explore further in future works.

---

> > ### Comment · Reviewer_e8kL · 2022-08-09
> > **Increased score**
> >
> > I would like to thank the authors for an incredibly thorough response, addressing all my concerns with the paper.  I have increased the score to 7.
> >
> > As a side note, the color palette adjustment helps tremendously.  I was unable to perceive the pattern differences in Fig 4, e.g., whereas they are quite obvious to me now when rendered in the new palette.

---

### Author Response · Authors · 2022-08-02
**Revision changes summary**

We would like to thank the reviewers once again for their overall positive comments and feedback.

We just wanted to summarize the improvements made to the paper in the uploaded revision as a result of the reviewers comments:

- Updated figure 5 to visualize the mean and standard deviation of multiple seeds, and provided a significance p-value between the SONS and SOM model in the text (p<.001) (line 250).
- Changed the color palette of some of the figures to be more colorblind friendly.
- Slightly expanded some sections to improve clarity (please see individual responses).
- Fixed some grammatical issues

We hope these changes and the given responses will have addressed most of the concerns raised.

---

### Meta-Review · Area_Chair_onNN · 2022-08-26

**Recommendation:** Accept
**Confidence:** Certain

**Metareview:**

Novel and sound contribution.

**Award:**

No

---

### Decision · Program_Chairs · 2022-09-14

Accept